# Treatment of Pediatric Inflammatory Myofibroblastic Tumor: The Experience from China Children’s Medical Center

**DOI:** 10.3390/children9030307

**Published:** 2022-02-24

**Authors:** Youhong Dong, Kashif Rafiq Zahid, Yidi Han, Pengchao Hu, Dongdong Zhang

**Affiliations:** 1Department of Oncology, Xiangyang No. 1 People’s Hospital, Hubei University of Medicine, Xiangyang 441000, China; dongyouhong2005@163.com (Y.D.); yangtian0415@whu.edu.cn (Y.H.); 2Department of Hematology, Zhongnan Hospital Affiliated to Wuhan University, No. 169 Donghu Road, Wuhan 430071, China; kashif_rafiq_zahid@yahoo.com; 3Shenzhen Key Laboratory of Microbial Genetic Engineering, College of Life Science and Oceanography, Shenzhen University, Shenzhen 518000, China; 4Carson International Cancer Center, School of Medicine, Shenzhen University, Shenzhen 518000, China; 5Xinhua Hospital Affiliated to Shanghai Jiao Tong University School of Medicine, Shanghai 200000, China

**Keywords:** inflammatory myofibroblastic tumor, China Children’s Medical Center, risk stratification, anaplastic lymphoma kinase

## Abstract

Background: Inflammatory myofibroblastic tumor (IMT) is a rare mesenchymal tumor with intermediate malignancy that tends to affect children primarily. To date, no standardized therapies exist for the treatment of IMT. This study aimed to share experience from China Children’s Medical Center for the explorative treatment of IMT. Methods: Patients with newly diagnosed IMT between January 2013 and December 2018 were included. Patients were grouped according to surgical margins and Intergroup Rhabdomyosarcoma Study Group (IRSG) staging. The clinical characteristic, therapeutic schedules, treatment response and clinical outcome were described. Results: Six patients were enrolled in this study, including two boys and four girls, with a median age of 57 months (range 10–148 months). Among them, five patients were anaplastic lymphoma kinase positive. Four patients achieved complete remission and two patients attained partial remission after treatment with this protocol. All patients were alive after a median follow-up of 4 years (range 3–7 years). The most common treatment-related adverse reaction was myelosuppression. Conclusion: In this study, we demonstrated that IMT has a good prognosis and the treatment selected according to risk stratification was effective and feasible.

## 1. Introduction

An inflammatory myofibroblastic tumor (IMT), originally known as an inflammatory pseudotumor, has the potential of recurrence and aggressive behavior [1]. The malignancy is also called by some other terms such as pseudosarcomatous myofibroblastic proliferation or inflammatory myofibrohistiocytic proliferation, which involves the myofibroblastic spindle cells and the infiltration of inflammatory lymphoplasma cells and eosinophils. In addition to its common occurrence in the lungs, IMT can also occur in various extrapulmonary sites including colon, genital tract, spleen, and orbital region [1,2]. About 30% of the patients with IMT may have clinical manifestations of inflammation symptoms.

IMT is defined as an intermediate soft tissue tumor by the World Health Organization, as it can be locally invasive and reoccur [3]. Multifocal disease and distant metastases are infrequent in IMT. Nevertheless, epithelioid inflammatory myofibroblastic sarcoma (EIMS), a unique subtype of IMT, has a relatively high incidence of local recurrence and distant metastases [4]. Surgery remains the mainstay of treatment for localized IMT, whereas systemic therapy is applicable in advanced disease or inoperable tumor sites. Conventional therapies include high-dose corticosteroids, non-steroidal anti-inflammatory drugs, cyclosporine A, vinblastine/methotrexate or doxorubicin- based chemotherapy [5,6], and local radiotherapy. As activating rearrangements of the anaplastic lymphoma kinase (*ALK*) gene were identified in 50% of cases, the targeted inhibition of ALK is a promising treatment option for IMT [7]. Other actionable genomic alterations, including ROS1, RET, NTRK, and PDGFRβ infusion, are also potential therapeutic targets [8]. Although various approaches have witnessed huge progress in the treatment of IMT, the development of standardized systemic therapy and effective clinical management approaches is urgently required. China Children’s Medical Center (CCMC) was the first to perform risk stratification for IMT. We demonstrated treatment diversity depending on risk grouping in this retrospective study.

## 2. Patients and Methods

### 2.1. Patient

Patients aged less than 18 years, newly diagnosed with IMT and previously untreated, were selected for the study from January 2013 to December 2018. The diagnosis of IMT was based on clinical presentation, radiology, conventional histology and immunohistology, and molecular pathology. The *ALK* status was determined by immunohistochemical evaluations. The clinical data were collected from the medical records. This study was approved by the Ethics and Scientific Committee of Hubei University of Medicine with approval number XH2021004. The patients’ parents gave written informed consent before the therapy was given.

### 2.2. Treatment

Table 1 lists the detailed group criteria and treatment regimens of our study. Patients in Group I received surgery only at the time of diagnosis. Patients classified into Groups II and Group III received two to four cycles of vincristine, doxorubicin, and cyclophosphamide (VDC) regimen, followed by surgery. Intensive low-dose vincristine plus methotrexate (MTX-V) regimen was adopted for patients in Group IV. The MTX-V regimen was also selected for patients in Group III with localized lesions at an inoperable location. Local radiotherapy was used alternatively in patients with distant metastases or macroscopic-positive resection. Topotecan plus cyclophosphamide (TC) regimen was selected as the second-line chemotherapy for relapsed patients. Moreover, when a patient presented with a positive ALK status, crizotinib was also recommended and orally administered every day for 1 year.

## 3. Treatment Response and Toxicity

A modified Response Evaluation Criteria in Solid Tumors (RECIST) was used to assess treatment response [9]. Complete remission (CR) was defined as the absence of all lesions for more than 4 weeks. Partial remission (PR) was characterized as a more than 64% decrease in the size of primary tumor and more than 30% reduction in metastatic lesions, with no new metastatic lesions. Progressive disease (PD) was defined as a more than 40% increase in the size of primary tumor or the appearance of new lesions. Stable disease was between PR and PD.

The necessary examination, including hematological and biochemistry tests, electrocardiogram, and electrocardiograph, was completed before every cycle of chemotherapy. Treatment-related adverse effects were graded according to the National Cancer Institute Common Terminology Criteria for Adverse Events version 4.0 [10].

## 4. Results

### 4.1. Patients

From January 2013 to December 2018, six patients newly diagnosed with IMT were enrolled in our study, including two boys and four girls. The median onset age was 57 months (range 10–148). Five patients demonstrated the activation of ALK. One patient had the EIMS subtype. The clinical and biological characteristics of all cases are shown in Table 2.

### 4.2. Treatment and Toxicity

The chemotherapy regimens used in this study are shown in Table 3. Patients in Group I underwent initial complete resection without any postoperative adjuvant treatment. Patients in Groups II-III underwent R1/R2 surgery, then received 2–4 cycles of adjuvant chemotherapy. Systemic chemotherapy was the main treatment for patients in Group IV. Radiotherapy could be considered for patients with respiratory distress syndrome, who did not show rapid response to chemotherapy or who could not tolerate chemotherapy. Only one patient received radiation after incomplete surgical resection at the prescribed dose of 24 Gy (5 × 1.8 Gy fractions per week).

The common adverse reactions were myelosuppression and gastrointestinal reaction. Three patients experienced febrile neutropenia. Four patients suffered from vomiting and diarrhea. One patient in Group IV developed numbness due to the long-term use of vinorelbine. The discomfort was alleviated by symptomatic treatment. Overall, the adverse reactions were tolerable and manageable.

### 4.3. Treatment Response and Clinical Outcome

All of the patients were alive during a follow-up period of at least 3 years. Patients in Group I and Patients in Group II-III who received adjuvant chemotherapy achieved CR with no reports of recurrence. One patient in Group II with *ALK* positive received radiotherapy after surgery, without adjuvant chemotherapy, and experienced systemic relapses twice. She achieved the first remission after treatment with MTX-V regimen combined with crizotinib. A second prolonged remission was achieved when the patient was treated with second-line chemotherapy (topotecan plus cyclophosphamide) combined with crizotinib. Of the two patients in Group IV, one patient achieved CR and another patient achieved PR. 

## 5. Discussion

Although IMT was first described in 1937, its etiology remains unclear or unknown. *ALK* was recognized as an oncogenic driver in lymphoma and some solid tumors [11,12]. Since all patients with EIMS and 50% of the patients with IMT patients present with the activation of *ALK*, there is no doubt that IMT is a true neoplasm rather than a reaction to inflammation. Therefore, active treatment is necessary. 

The dominance of surgery has been well-established in resectable lesions. The question was whether postoperative chemotherapy was required. Several studies showed that a small proportion of patients experienced local recurrence within 3 months after surgical resection without adjuvant therapy [7], which might provide evidence for the need for adjuvant postoperative chemotherapy. According to the experience from the European pediatric Soft Tissue Sarcoma Study Group (EpSSG), the “wait and watch” strategy could be adopted for patients in Group I and systemic chemotherapy could be administered to patients in Group III; whether adjuvant chemotherapy was necessary for patients in Group II was still controversial [13]. Which chemotherapy had a high degree of activity in IMT was another question that needed to be addressed. One European retrospective study demonstrated that anthracycline-based regimens remained the front-line standard treatment of IMT [6]. However, a multicenter retrospective case-series analysis indicated that the MTX-V regimen had a similar object remission rate (ORR) compared with the anthracycline-based regimens in advanced IMT (54% versus 48%); more importantly, the MTX-V regimen had more prolonged disease control [14]. In our study, patients in Group II and III were treated with VDC regimen and MTX-V regimen was given to patients in Group IV and those with challenging primary sites that were inoperable. This indicated that some pediatric patients might have been overtreated in this retrospective study. Considering the cardiotoxicity of anthracycline and the risk of secondary tumor caused by alkylating agents, we are more likely to recommend the MTX-V regimen as the first-line treatment for advanced IMT in the future.

Whether patients with a positive resection margin can be benefit from radiation is still uncertain. In our study, patient 3 (the age at diagnosis was 123 months) with R1 section in the primary tumor site was given local external irradiation without chemotherapy. She did not experience local recurrence after the 4-year follow-up, but new lesions were found in other sites. Considering the long-term sequelae such as growth retardation caused by radiation in very young children, we held the opinion that radiotherapy was not recommended for patients aged less than 3 years, no matter which group they were classified in. For patients aged more than 3 years in Group IV, radiation could be used as an alternative therapy because IMT had a good prognosis, but it must be individualized.

ALK inhibition proved to be highly effective and was recommended as a first-line treatment for IMT considered *ALK* positive by the Children’s Oncology Group [15,16]. However, ALK inhibitors may be unavailable for patients in developing countries because of the high cost. Moreover, many questions concerning the treatment duration, the mechanism of drug resistance, and the management of drug resistance remain unanswered. Therefore, crizotinib was administered as a second-line treatment for the only refractory patient in our study. A study showed that the acquired drug resistance to crizotinib resulting from the ALK (G1269A) mutation was reversible by the next-generation ALK inhibitor [17,18]. Rituximab might be beneficial for ALK-negative recurrent IMT [19].

We could not recruit enough patients in a single institute because of the rarity of IMT and the young age at onset; therefore, it was difficult for us to evaluate the effectiveness of this protocol based on this risk stratification, and we did not analyze the long-term survival. So far, no consistent guideline recommendation exists for clinicians because most studies on IMT are sporadic case reports. We formulated a treatment strategy for IMT based on the experience from EpSSG and CCMC to help the clinicians (Figure 1). However, we realized that further research with both a larger sample and multiple centers is still needed.

In conclusion, we formulated the treatment strategy for IMT according to risk stratification. Surgery was still the mainstream treatment. A “wait and watch” strategy was applied for patients in Group I. Whether adjuvant treatment was needed after surgery for patients in Group II should be individualized. Systemic chemotherapy should be given to patients in Group III after surgical resection or biopsy alone. Systemic chemotherapy was the main treatment for patients in Group IV, *ALK* inhibitors might be an alternative choice for patients who are *ALK* positive. Even so, further studies with a larger patient population are needed to evaluate the treatment efficiency.

## Figures and Tables

**Figure 1 children-09-00307-f001:**
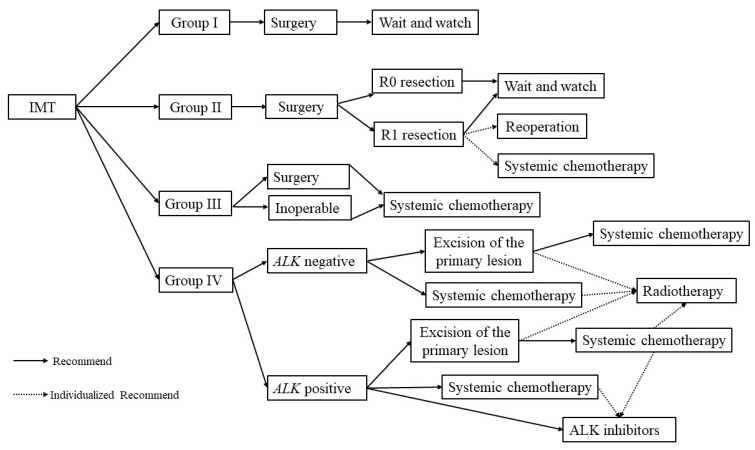
The treatment strategy for IMT.

**Table 1 children-09-00307-t001:** Clinical group and treatment in China Children’s Medical Center (CCMC).

Group	Definition	Treatment
I	Localized lesions with R0 resection without regional lymph node metastasis	Surgery
II	Localized lesions with R1 resection or regional lymph nodal spread	Surgery plus 2 cycles of doxorubicin, vincristine and cyclophosphamide
III	Localized lesions with R2 resection or biopsy alone (including localized lesions at inoperable location)	Surgery plus 4 cycles of doxorubicin, vincristine and cyclophosphamide
IV	Distant metastases at onset	Methotrexate plus vinorelbine every week for 1 year, radiotherapy or ALK inhibitor were individual recommended.

Note: R0, negative resection margin; R1, microscopic positive resection margin; R2, macroscopic positive resection.

**Table 2 children-09-00307-t002:** Clinical characteristics, treatment-related toxicities and treatment response of 6 patients with inflammatory myofibroblastic tumor (IMT).

Patient	Sex	Age (Months)	Tumor Site	Metastatic Site	Histopathology (ALK Status)	Tumor Size	Group	Treatment and Response	Toxicities
1	Male	10	Mesentery	None	IMT (Positive)	6.4 × 4.6 × 7.0 cm	II	Primary R1 resection, followed by two cycles of VDC regimen. CR was achieved 3 months after initial diagnosis and still in CR after 7 years.	Grade IV myelosuppression, febrile neutropenia
2	Female	11	Mesentery	None	EIMS (Positive)	15.0 × 12.0 × 8.0 cm	III	Primary R2 resection, followed by four cycles of VDC regimen. CR was achieved 5 months after initial diagnosis and still in CR after 3 years.	Grade II myelosuppression and vomiting
3	Female	123	Left thigh	None	IMT (Positive)	4.0 × 3.4 × 7.0 cm	II	Primary R1 resection, followed by local radiotherapy (24 Gy). New lesions (Lung, right thigh) were appeared 11 months after initial diagnosis.	None
							IV	Systemic chemotherapy: MTX-V regimen was given for 1 year, followed by crizotinib (2 × 125 mg/d). New lesions (Right thigh, acetabulum) were appeared 30 months after initial diagnosis.	Grade II vomit, Grade III numbness
							IV	Second-line treatment: four cycles of TC regimen was given, followed by crizotinib (2 × 125 mg/d). PR was achieved 4 months after the completion of treatment and still in PR after 63 months.	Grade II vomit and diarrhea, Grade IV myelosuppression, febrile neutropenia
4	Female	148	Shoulder-back	Lung	IMT (Positive)	6.0 × 4.2 × 3.0 cm	IV	R0 resection of the primary lesion, followed by MTX-V regimen for 1 year. PR was achieved 14 months after initial diagnosis and still in PR after 4 years.	Grade IV myelosuppression, febrile neutropenia, Grade II vomiting and diarrhea
5	Male	13	Scrotum	None	IMT (Negative)	2.3 × 1.8 × 0.9 cm	I	R0 resection, followed by wait and watch. CR was achieved 2 months after the initial diagnosis and still in CR after 3 years.	None
6	Female	39	Abdomen	Lymth node	IMT (Positive)	4.0 × 4.0 × 5.0 cm	IV	R0 resection of the primary lesion, followed by MTX-V regimen for 1 year. PR was achieved 13 months after initial diagnosis and still in PR after 4 years.	Grade II vomiting and diarrhea

Note: CR, complete remission; PR, partial remission.

**Table 3 children-09-00307-t003:** The vincristine, doxorubicin, and cyclophosphamide (VDC) regimen, vincristine plus methotrexate (MTX-V) regimen and topotecan plus cyclophosphamide (TC) regimen.

Agents	Does	Route	Time
VDC			
Vincristine	1.5 mg/m^2^ (D_max_ = 2 mg) (<12 kg: 0.5 mg/kg)	Push	D1
Doxorubicin	30 mg/m^2^ (<12 kg: 1.0 mg/kg)	IV	D1–2
Cyclophosphamide	1.2 g/m^2^ (<12 kg: 40 mg/kg)	IV	D1
MTX-V			
Methotrexate	30 mg/m^2^	IV or orally	D1
Vinorelbine	20 mg/m^2^	IV	D1
TC			
Topotecan	1.2 mg/m^2^	IV	D1–5
Cyclophosphamide	400 mg/m^2^ (<12 kg: 13.3 mg/kg)	IV	D1–5

Note: VDC regimen and TC regimen were given at the interval of 21 days. MTX-V was given weekly at the first 6 months and fortnightly for the subsequent 6 months.

## Data Availability

The clinical data supporting the conclusions of this manuscript will be made available by the authors.

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
