# Peer review of "Treatment of Pediatric Inflammatory Myofibroblastic Tumor: The Experience from China Children’s Medical Center"

_children, 2022, doi:10.3390/children9030307_

Round 1

Reviewer 1 Report

The authors of the paper presented 6 cases of IMT. They described clinical symptoms, stages and therapeutic procedures. Based on their own observations, they proposed the treatment strategy for IMT.

The advantage of the study is the presentation of collective cases, which is not common due to the rarity of this diagnosis. A graphical the treatment strategy for IMT may be useful in clinical practice, but the proposed methods of therapy are well known.

Author Response

We appreciate the insightful comments raised by the reviewers, and we have revised our manuscript accordingly. The detailed reply to the Reviewer's comments are attached below.

Reviewer 2 Report

The authors describe a case series or a retrospective analysis of a case series of children with IMT. 

This is of interest to paediatric oncologists, as IMT is a very rare condition, only small numbers of cases have been reported so far, and there are no unanimously accepted treatment guidelines.

There are some important issues, however, which the authors should address before publication:

Patients and Methods, LL 59 ff, andTable 1:

For me, it is not quite clear from this section, what the nature of the study exactly was. Was this just a retrospective case file analysis, or was the treatment part of a prospectively planned treatment study, including the clinical grouping as described in Table 1? The authors should specify this, please. If this was a prospectively planned study, the Methods section needs to include some details on this like a study plan, and which analyses had originally be planned. If, however, the data reported here are merely results from a retrospective case file analysis, the authors should clearly state this. In this context, it is of interest, whether the clinical grouping as outlined in Table 1 was planned ahead of the patient recruitment, and then applied before treatment, or whether this grouping was only done retrospectively, i.e. after treatment, and was done only for this report.

Diagnosis, LL 62 ff, reads:  “The diagnosis of IMT 62 was based on clinical presentation, radiology, and molecular pathology.“  I assume, conventional histology and immunohistology were used as well?

“Treatment” section, LL 68 ff, and Table 2:

I am not quite sure if I understand all treatment details. This, however, is very important, as the treatment success is the main topic here. As only six patients are included, it might be helpful to expand the treatment and outcome of patients from Table 2 into a brief narrative for each patient, e.g. “Patient 1, primary R1 resection,  followed by 2 cycles of VDC, resulting in CR xy months after initial diagnosis. Patient is still in CR after 7 years.” (This, at least, is what I assume patient one’s story goes like.) Especially, the course in the somewhat complex patient 3 is not entirely clear to me: Has she received initial R1 resection, radiotherapy, and developed lung metastases 11 months thereafter (or 11 months after diagnosis?),  was then treated by MTX-V, crizotinib, and again developed new metastases in the acetabulum 30 months later, which were not operated on, but lead to treating her with TC and crizotinib again, which made her achieve PR, which at the time of analysis had lasted for 33 months? Maybe the authors could give such details in the text on all patients.

In patients 4 and 6, the surgical procedure is not described as R0, R1, or R2, but rather is given as “Excision”. Are there no data on surgical margins in these two cases?

Line 87: “Stable disease was between PR and progressive disease.” The authors should define “progressive disease”.

Line 105-106: “Surgery was performed only on patients in Group I.” What does this mean? There is only one patient in group I, and he received complete surgery. Is that what it means? If so, the authors might rather state, that only in one patient complete surgery was achieved.

Line 106: “Patients in Groups II-III received adjuvant chemotherapy, followed by surgery.” I had the impression, that patients received surgery before chemotherapy. Please clarify!

LL122 ff: “All of the 122 patients in Groups I-III achieved CR after adjuvant chemotherapy with no reports of recurrence. Of the three patients in Group IV, one patient achieved CR and two other patients achieved PR.”

This information seems not entirely consistent with the information given in Table 2: Patient 3, being in group II in the beginning, experienced systemic relapses twice, before reaching PR. So yes, she did not have local recurrence, but systemic metastases instead. Thus, the authors should rephrase this to make the course clearer, I think. As for patients in Group IV, at initial presentation there were not 3, but only 2 patients in Group IV, as far as I can see from Table 2 (who indeed achieved 1 CR and 1 PR, respectively).

Round 2

Reviewer 2 Report

Thank you for re-editing the manuscript, which is much clearer now.

I think this may be published now.